# *Clostridioides difficile* colonization amplification despite limited in-hospital transmission: A modeling study

**Daniel De-la-Rosa-Martinez**[1]*, **Travis C. Porco**[1,2], **Ashley Hazel**[1], **Xinran Liu**[3], **Karim Khader**[4], **Seth Blumberg**[1]

**1** Francis I. Proctor Foundation, University of California San Francisco, San Francisco, California, United States of America, **2** Department of Epidemiology and Biostatistics, University of California San Francisco, San Francisco, California, United States of America, **3** Department of Medicine, University of California San Francisco, San Francisco, California, United States of America, **4** Department of Internal Medicine, University of Utah, Salt Lake City, Utah, United States of America

* daniel.delarosamartinez@ucsf.edu, dan.delarosam@gmail.com

## Abstract

### Background

Although healthcare-associated transmission of *Clostridioides difficile* is a recognized public health concern, community-onset infections represent an important component of the overall disease burden. This paradox likely reflects the underappreciated interplay between these settings. We aimed to quantify in-hospital transmission and the hospital's contribution to community colonization by estimating the intrinsic reproduction number ($R_i$) and introducing the colonization amplification index ($A_i$), defined as the ratio of colonized patients at discharge to those at admission. Given the potential contribution of external cases, we also evaluated interventions targeting asymptomatic carriers at admission to reduce disease burden.

### Methods and findings

We developed a compartmental model informed by data from UCSF Medical Center to capture *C. difficile* transmission dynamics among symptomatic and asymptomatic patients. Across simulations, the median $R_i$ was 0.61 (Q1–Q3: 0.53, 0.71), consistently indicating limited sustained in-hospital transmission ($R_i < 1$). In contrast, $A_i$ was 1.9 (Q1–Q3: 1.7, 2.1), suggesting substantial amplification of colonization during hospital stay. Amplification of colonization persisted in sensitivity analyses, with $R_i$ exceeding 1 under boundary values of low colonization prevalence at admission, a low proportion of colonized patients progressing to symptomatic disease, and prolonged incubation periods. Redefining healthcare-associated *C. difficile* infection (CDI) using thresholds from ≥1-day post-admission instead of the standard threshold of >3 days post-admission increased $A_i$ and $R_i$ by 11% and 21%, respectively. A threshold of ≥5 days post-admission reduced these metrics by 5% and 8%,

**Data availability statement:** Code and data necessary to reproduce the presented analyses are openly available from the following repository: https://github.com/danieldelarosam/C.-difficile-Modeling and on Zenodo: https://doi.org/10.5281/zenodo.19392691.

**Funding:** This study was supported by the Centers for Disease Control and Prevention [U01CK000590 to DD, TP, AH, XL, and SB] and [1U01CK000585-01 to KK] (https://www.cdc.gov) as part of the Modeling Infectious Diseases in Healthcare Network & NIH NIGMS (https://www.nigms.nih.gov) [R35GM147702 to DD and SB]. The funders had no role in study design, data collection and analysis, decision to publish, or preparation of the manuscript.

**Competing interests:** The authors have declared that no competing interests exist.

**Abbreviations:** Ai, amplification index; CA-CDI, community-associated C. *difficile infection*; CDI, *Clostridioides difficile* infection; HCA-CDI, healthcare-associated *C. difficile* infection; ICUs, intensive care units; PRCC, Partial rank correlation coefficients.

respectively. Interventions targeting asymptomatic carriers through contact precautions and/or prophylactic treatment reduced both $A_i$ and CDI incidence, with combined interventions yielding the greatest reductions, followed by contact precautions alone. Our main limitation stems from uncertainty in some parameters describing the disease's natural history, particularly colonization prevalence at admission, progression to symptomatic disease, and the incubation period, which may affect the precision of our estimates despite sensitivity analyses.

## Conclusions

Our findings indicate that in most potential scenarios, in-hospital transmission of *C. difficile* is limited ($R_i < 1$) and likely sustained by continuous importation of cases from the community. Nevertheless, hospitalization amplifies colonization ($A_i > 1$), which potentially contributes to community transmission. These results underscore the importance of interventions addressing asymptomatic carriers, a currently overlooked source of spread. Our study highlights the need to broaden metrics beyond $R_i$ to capture hospitals' contribution to the *C. difficile* burden. Future infection control strategies should address colonization dynamics at admission and potentially at discharge to mitigate transmission and reduce the overall burden of *C. difficile*.

---

## Author summary

### Why was this study done?

- *Clostridioides difficile* is an important healthcare-associated pathogen due to its high morbidity and capacity for patient-to-patient transmission; its hospital burden reflects both within-hospital spread and the admission of carriers with or without symptoms.

- Most studies focus on symptomatic infections and pay limited attention to the interaction between community importation by asymptomatic carriers, in-hospital transmission, and reintroduction back into the community when asymptomatic patients are discharged.

- Since asymptomatic carriers play a significant role in transmission, it is necessary to evaluate the potential impact of preventive measures targeting this population.

### What did the researchers do and find?

- We developed a mathematical model of *C. difficile* transmission in a hospital to characterize the burden of disease and evaluate interventions that target asymptomatic patients.

PLOS Medicine

- Across most reasonable choices of transmission parameters that are consistent with clinical data and published literature, transmission within hospitals was self-limiting. However, when compared to patients admitted to the hospital, there was consistent amplification of infection amongst discharged patients.

- Interventions targeting asymptomatic carriers at admission reduced both colonization at discharge and the number of symptomatic infections.

## What do these findings mean?

- Even when in-hospital transmission is limited, hospitals may still increase the number of colonized individuals who return to the community after discharge, potentially influencing community transmission.

- Measures focused only on diagnosed infections miss opportunities for infection control.

- Our quantitative estimates of within-hospital transmissibility and the effectiveness of control depend on our specific model assumptions and choices of parameter values. However, the overall findings that hospitals tend to amplify *C. difficile* infection and controlling asymptomatic, within-hospital spread decreases community exposure are expected to be robust.

## Introduction

Mathematical models of *Clostridioides difficile* transmission have traditionally focused on the within-hospital dynamics of healthcare-associated infection and the impact of control interventions [1,2]. Relatively less attention has been paid to the interplay between community- and hospital-based transmission dynamics, despite growing evidence that asymptomatic carriers and community-imported cases play a critical role in sustaining transmission [1,3–6]. Colonized patients discharged from healthcare settings can potentially introduce the pathogen into the community, while individuals colonized or infected outside the hospital can reintroduce *C. difficile* into healthcare facilities, creating a feedback loop that may contribute to infection persistence [7]. This study explores this relationship by developing a compartmental model informed by clinical data from a university-affiliated medical center.

Traditional transmission models often rely on the reproduction number, defined as the average number of secondary cases generated per index case, to quantify in-hospital transmission. However, this metric typically assumes a closed population and does not account for community importation of cases or transmission that occurs after discharge, thus limiting its ability to reflect the role of healthcare institutions in sustaining broader community transmission [8–11]. To address this, we introduce the amplification index ($A_i$) as the ratio of colonized patients at discharge to those at admission. This index serves as a complementary metric to the reproduction number, capturing both in-hospital transmission and the hospital's contribution to community spread.

A second challenge of quantifying transmission relates to how symptomatic infections are classified as community-associated (CA-CDI) and healthcare-associated *C. difficile* infection (HCA-CDI). Traditionally, this classification relies on heuristic, time-based definitions that assign the infection's origin based on the timing of symptom onset relative to admission [12]. The higher the proportion classified as CA-CDI, the lower the estimate of healthcare-associated transmission. Here, we assess how sensitive transmission estimates are to these thresholds.

Our observation that the community-healthcare interface heavily influences the burden of HCA-CDI motivates alternative approaches to assessing within-hospital transmission and evaluating control strategies focused on the importation of asymptomatic carriers of *C. difficile.* In this study, we develop a mathematical model to estimate hospital transmission and assess the sensitivity of transmission metrics to case definition thresholds. We also evaluate the potential impact of interventions for asymptomatic carriers at admission, including screening and contact precautions and/or prophylactic

decolonization. Our focus on interventions addressing asymptomatic transmission fills an important research gap, as current infection prevention strategies focus almost exclusively on symptomatic cases, despite substantial evidence that asymptomatic individuals can act as reservoirs for onward transmission in healthcare facilities and the community [12–16].

## Methods

While the term *infected* can be used in the literature to denote the presence of bacteria in the body regardless of clinical symptoms, in this work, we use the term specifically to refer to individuals who exhibit symptomatic infection and test positive for *C. difficile* by PCR. While this definition may include some patients with diarrhea unrelated to *C. difficile*, it prioritizes more robust evaluation of transmission.

### Model description

We model *C. difficile* hospital transmission using a deterministic compartmental framework that includes eight compartments: 1) Nonsusceptible (R) patients whose gut microbiota has not been disrupted by antibiotics. 2) Susceptible (S) individuals with disrupted microbiota due to antibiotic exposure, who are at risk of *C. difficile* colonization. 3) Short-term asymptomatic carriers (E) or patients colonized with toxigenic *C. difficile*, who may progress to symptomatic infection during their hospital stay. 4) Long-term asymptomatic carriers (C) or patients with persistent colonization who will remain asymptomatic during their hospital stay. 5) Infected (I) individuals or patients with symptomatic *C. difficile* infection, confirmed by PCR. 6) Diagnosed infected (A) individuals who are on contact precautions and antibiotic treatment. Additionally, to explore the potential benefits of contact precautions on asymptomatic carriers, we include two compartments: (7) Diagnosed and isolated asymptomatic carriers type E ($K_1$), and (8) type C ($K_2$). As part of the intervention, these carriers may be treated and either move to the susceptible compartment or remain colonized if treatment is unsuccessful or not administered. Movement rates between compartments are illustrated in Fig 1.

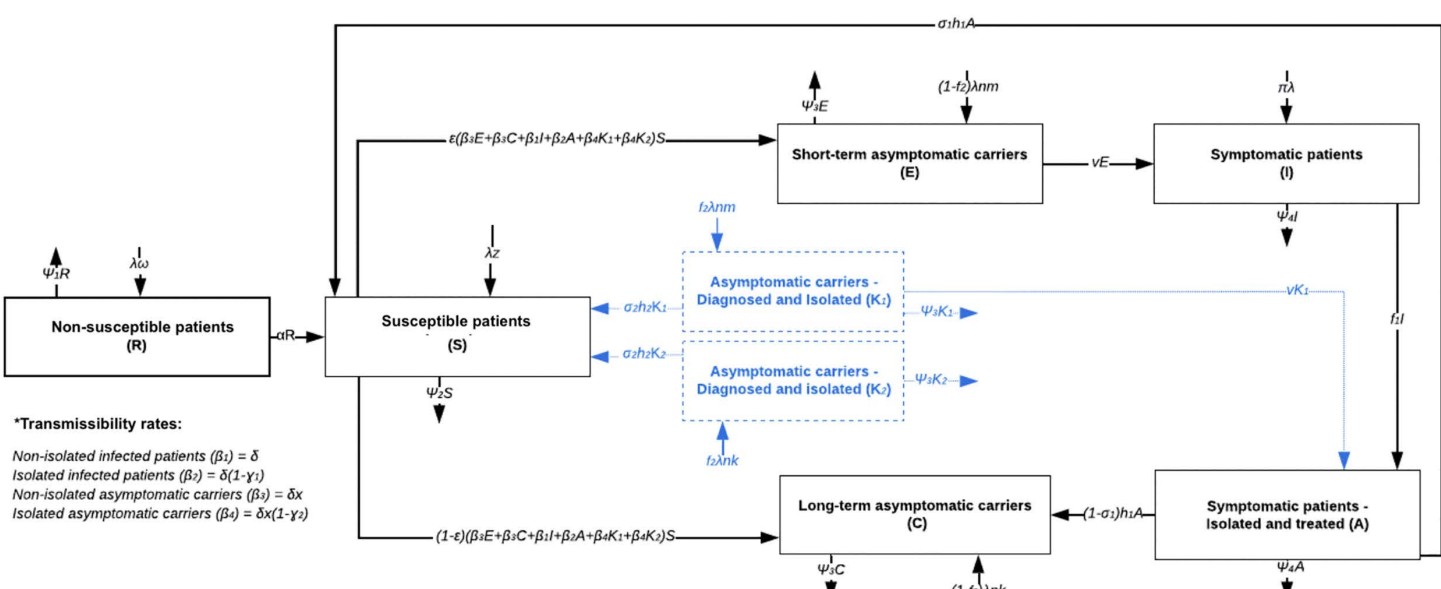

**Fig 1. Compartmental model workflow of disease dynamics.** Compartments and arrows in black depict the infection dynamics under standard conditions or the baseline transmission model. In contrast, blue-dotted compartments ($K_{1-2}$) and arrows represent hypothetical compartments based on control interventions for asymptomatic carriers, including contact precautions and decolonization.

## Rates, assumptions, and parameters

**1.1  Movement from nonsusceptible (R) to susceptible (S) compartment.**  Antibiotic use affects the transition from the nonsusceptible to the susceptible population. These agents disrupt the gut microbiome and increase vulnerability to *C. difficile*. Our model uses the α parameter to capture exposure to high-risk antibiotic classes during hospitalization, including quinolones, cephalosporins, sulfonamides, penicillins, macrolides, aminoglycosides, and lincosamides. Therefore, the transition from the R→S compartment is denoted as αR.

**1.2 Force of infection of colonized and infected individuals.**  Susceptible patients can acquire infection from both symptomatic and asymptomatic carriers. The transmission rate of infected individuals is denoted as $\delta$. We assume that this rate is reduced by contact precautions, which are typically imperfect due to variable compliance and intrinsic limitations. We denote the overall effectiveness of contact precautions for infected individuals as $\gamma_1$. Therefore, the transmissibility rate of infected individuals under contact precautions (compartment A) is calculated as the product of the transmission rate of the infected population ($\delta$) and one minus contact precaution effectiveness ($\gamma_1$). We use the terms $\beta_1$ and $\beta_2$, where:

$$\beta_1 = \delta \text{ or transmission rate of nonisolated infected patients,}$$

$$\beta_2 = \delta(1 - \gamma_1) \text{ or transmission rate of isolated infected patients.}$$

Based on previous evidence, we assume that asymptomatic patients can transmit the infection at a lower rate than patients manifesting active symptoms [14]. In our model, this reduction in transmissibility is indicated by the parameter *x*, which represents the relative transmissibility of the asymptomatic carrier population. Similar to the infected population, the transmissibility of the asymptomatic carrier population can be modified by contact precautions. Therefore, the transmission rate for nonisolated asymptomatic carriers is defined as $\delta x$, while the asymptomatic carrier population on contact precaution has a transmission rate denoted as $\delta x(1-\gamma_2)$, which depends on the effectiveness of contact precaution on asymptomatic carriers, represented as $\gamma_2$. We define:

$$\beta_3 = \delta x \text{ or transmission rate of nonisolated asymptomatic carriers.}$$

$$\beta_4 = \delta x (1 - \gamma_2) \text{ or transmission rate of isolated asymptomatic carriers.}$$

**1.3 Movement from susceptible (S) to colonized (E or C) compartments.**  Susceptible patients can transition into one of two distinct types of asymptomatic carriers: those in the incubation period who may develop symptomatic disease (E compartment) during hospitalization and those who remain asymptomatic throughout their hospital stay (C compartment). Although the factors influencing this progression rate are not fully understood, they may be related to immunological response, microbiome diversity, or *C. difficile* virulence characteristics [5]. Accordingly, the relative movement into each compartment is modeled by the parameter ε, representing the proportion of newly colonized patients who may develop symptomatic disease and enter compartment E.

**1.4 Movement from the pre-symptomatic (E) to the infected compartment (I) and the treated/isolated compartment (A).**  We used the parameter ν to represent the rate of symptom development, calculated as the inverse of the days from bacterium acquisition to onset of symptoms. Thus, the transition rate from the E→I compartment is represented as νE. Once patients develop symptomatic disease, they are diagnosed at a rate of $f_1$ and move from the I→A compartment.

Patients complete treatment at a rate of $h_1$, the inverse duration of antibiotics recommended for infection therapy. Following treatment, a proportion $\sigma_1$ of patients fully clear the bacteria and transition to the S compartment, reflecting treatment effectiveness. In contrast, the remaining proportion, $1 - \sigma_1$, does not clear the infection and remains colonized, moving to the C compartment. Therefore, $\sigma_1 h_1$ denotes the transition rate from the A→S, and $(1-\sigma_1)h_1$ represents the

movement from the A→C compartment. Although some individuals who do not fully clear the bacteria may experience relapse of infection, such episodes typically occur after discharge and were not included in this model.

**1.5 Admission rate and community sources of nonsusceptible, susceptible, asymptomatic, and infected patients.** The overall admission rate is represented by $\lambda$, derived from clinical data as the average daily number of admissions at the institution during the study period. The proportion of patients admitted from the community with symptomatic infection or asymptomatic colonization is denoted as $\pi$ and $n$, respectively. A fraction $m$ of asymptomatic carriers is assumed to be at risk for symptomatic infection during hospitalization and enter the E compartment. The remaining asymptomatic carriers, $1-m$ (denoted as $k$), stay asymptomatic and enter the C compartment. The proportion of patients, $z$, who received antibiotics before admission but did not carry *C. difficile* enter the S compartment. The proportion $(1 - (n + \pi + z))$, denoted as $\omega$, enters the R compartment.

**1.6 Simulated treatment and isolation of asymptomatic carriers.** We modeled active screening and isolation of asymptomatic carriers at admission through compartments $K_1$ and $K_2$. The parameter $f_2$ defines the proportion of carriers identified upon admission. As a result, $f_2\lambda nm$ and $f_2\lambda nk$ represent the rates at which patients initially assigned to E and C are instead moved to $K_1$ and $K_2$, respectively. The remaining carriers, those not identified, enter E and C at rates $(1 - f_2)\lambda nm$ and $(1 - f_2)\lambda nk$.

Patients in $K_1$ and $K_2$ are placed under contact precautions until discharge. If treatment is initiated, a fraction $\sigma_2$ will successfully clear the bacteria at a rate $h_2$, corresponding to the treatment duration. We model these patients as transitioning to the susceptible compartment (S) at rate $\sigma_2 h_2$. However, the actual proportion of them that transition to S is different from $\sigma_2$ since multiple outflow processes are being modeled. Individuals in $K_1$ remain at risk of developing symptomatic disease. They move to compartment A at a rate $\nu K_1$, where $\nu$ is the incubation rate.

**1.7 Discharge rates.** We estimate the discharge rates from compartments based on hospitalization length data. Nonsusceptible, susceptible, and asymptomatic carriers are discharged at rates $\Psi_1$, $\Psi_2$, and $\Psi_3$, respectively, each corresponding to the inverse of the average length of hospital stay. For the infected population, the discharge rate is denoted as $\Psi_4$, representing the inverse of the average hospital stay for patients diagnosed with CDI. We describe the movements among compartments by the following differential equations:

$$\frac{dR}{dt} = \lambda\omega - \Psi_1 R - \alpha R \tag{1}$$

$$\frac{dS}{dt} = \alpha R - \left(\beta_3 E + \beta_3 C + \beta_1 I + \beta_2 A + \beta_4 K_1 + \beta_4 K_2\right) S - \Psi_2 S + \sigma_2 h_2 K_1 + \sigma_2 h_2 K_2 + \sigma_1 h_1 A + \lambda z \tag{2}$$

$$\frac{dE}{dt} = \varepsilon \left(\beta_3 E + \beta_3 C + \beta_1 I + \beta_2 A + \beta_4 K_1 + \beta_4 K_2\right) S + (1 - f_2)\lambda nm - \nu E - \Psi_3 E \tag{3}$$

$$\frac{dC}{dt} = (1 - \varepsilon) \left(\beta_3 E + \beta_3 C + \beta_1 I + \beta_2 A + \beta_4 K_1 + \beta_4 K_2\right) S + (1 - f_2)\lambda nk - \Psi_3 C + (1 - \sigma_1)h_1 A \tag{4}$$

$$\frac{dK_1}{dt} = f_2\lambda nm - \sigma_2 h_2 K_1 - \Psi_3 K_1 - \nu K_1 \tag{5}$$

$$\frac{dK_2}{dt} = f_2\lambda nk - \sigma_2 h_2 K_2 - \Psi_3 K_2 \tag{6}$$

$$\frac{dI}{dt} = vE + \lambda\pi - \Psi_4 I - f_1 I \tag{7}$$

$$\frac{dA}{dt} = f_1 I - (1 - \sigma_1) h_1 A - \sigma_1 h_1 A - \Psi_4 A + vK_1 \tag{8}$$

## Model parameters and calibration

Model parameters were obtained from clinical data from the University of California, San Francisco Medical Center, a large tertiary-care, academic hospital, from November 1, 2019, to May 31, 2021. Parameters unavailable in institutional data were derived from previously published studies (Table 1) [8,12,14,17–22]. Due to the absence of identifiable patient information and use of retrospective data only, our institutional review board declared our study exempt from full review.

We implemented our compartmental model using the deSolve package in R (version 4.3.2). To incorporate uncertainty in parameter values, we conducted 1,000 simulations in which model parameters, except for the transmission rate, were independently sampled from uniform distributions within ±20% of their baseline estimates (Table 1). The transmission rate or δ was optimized in each simulation through numerical calibration by minimizing the absolute difference between the model-predicted cumulative incidence of CDI, quantified as the total number of individuals passing through compartment A, and the observed number of infections over the study period. Initial conditions for simulations were set at the disease-free equilibrium.

## Statistical analysis

**Estimation of the intrinsic reproduction number.** In our model, individuals with *C. difficile* can either be colonized upon hospital admission or acquire the organism during hospitalization. Both groups can contribute to onward transmission. However, to evaluate the potential for sustained in-hospital transmission independent of importation, we define the intrinsic reproduction number ($R_i$) as the average number of secondary cases caused by individuals who became colonized during their hospital stay. For each set of parameters, we calculate $R_i$ using the next-generation matrix method [23]. This approach decomposes the system of differential equations into two vector functions:

$\mathcal{F}(\boldsymbol{x})$: representing the rate of appearance of new infections in each infectious compartment

$\mathcal{V}(\boldsymbol{x})$: representing the transfer rate into and out of each infectious compartment due to all other transitions.

Let $c$ denote the vector of infectious compartments. For our model, we define:

$$c = [E,\ C,\ K_1, K_2,\ I,\ A]_T$$

We then compute the Jacobian matrices of $\mathcal{F}(c)$ and $\mathcal{V}(c)$ for the infectious compartments, with entries defined by the partial derivatives:

$$F_{ij} = \frac{\partial \mathcal{F}_i(c)}{\partial \boldsymbol{c}_j}, \quad V_{ij} = \frac{\partial \mathcal{V}_i(c)}{\partial \boldsymbol{c}_j},$$

$$\mathbf{F} = \begin{bmatrix} \varepsilon\beta_3 S' & \varepsilon\beta_3 S' & \varepsilon\beta_4 S' & \varepsilon\beta_4 S' & \varepsilon\beta_1 S' & \varepsilon\beta_2 S' \\ (1-\varepsilon)\beta_3 S' & (1-\varepsilon)\beta_3 S' & (1-\varepsilon)\beta_4 S' & (1-\varepsilon)\beta_4 S' & (1-\varepsilon)\beta_1 S' & (1-\varepsilon)\beta_2 S' \\ 0 & 0 & 0 & 0 & 0 & 0 \\ 0 & 0 & 0 & 0 & 0 & 0 \\ 0 & 0 & 0 & 0 & 0 & 0 \\ 0 & 0 & 0 & 0 & 0 & 0 \end{bmatrix}$$

**Table 1. Description of input parameters and values used in estimations.**

| Symbol | Parameter description (units) | Baseline values | Range | Source |
|---|---|---|---|---|
| $\delta$ | Transmission rate of nonisolated symptomatic patients (Days$^{-1}$) | 0.00072 | Calibrated with data | Calibrated with data |
| $\alpha$ | Rate of antibiotic use (Days$^{-1}$) | 0.064 | (0.051, 0.076) | Clinical data |
| $\Psi_{1-3}$ | Discharge rate of nonsusceptible patients, susceptible patients, and asymptomatic carriers (Days$^{-1}$) | 0.166 | (0.13, 0.2) | Clinical data |
| $\Psi_4$ | Discharge rate of symptomatic patients (Days$^{-1}$) | 0.083 | (0.06, 0.1) | Clinical data |
| $h_1$ | Bacterial clearance rate due to treatment for infected patients or treatment duration (Days$^{-1}$) | 0.100 | (0.08, 0.12) | [12] |
| $\lambda$ | Admission rate (Patients/day) | 76 | – | Clinical data |
| $n$ | Fraction of patients admitted who are asymptomatic carriers (Dless) | 0.08 | (0.06, 0.09) | [17,18] |
| $m$ | Fraction of admitted asymptomatic carriers to the E compartment (Dless) | 0.12 | (0.09, 0.14) | [18] |
| $k$ | Fraction of asymptomatic carriers admitted to the C compartment (Dless) | 1-m | – | – |
| $\pi$ | Fraction of patients admitted who have symptomatic infections (Dless) | 0.005 | (0.004, 0.006)[a] | Clinical data |
| $z$ | Fraction of patients admitted who are not carriers, but are susceptible to colonization (Dless) | 0.22 | (0.17, 0.26) | [8] |
| $x$ | Relative transmissibility for asymptomatic carriers (Dless) | 0.700 | (0.56, 0.84) | [14] |
| $\varepsilon$ | Fraction of new asymptomatic carriers who may develop symptomatic disease (Dless) | 0.130 | (0.10, 0.15) | [18] |
| $v$ | Progression rate to symptomatic disease (Days$^{-1}$) | 0.250 | (0.20, 0.30) | [12] |
| $\sigma_1$ | Effective fraction of symptomatic patients clearing bacteria following treatment (Dless) | 0.700 | (0.56, 0.84) | [19] |
| $f_1$ | Diagnosis rate of symptomatic patients (Days$^{-1}$) | 0.666 | (0.53, 0.80) | [22] |
| $\gamma_1$ | Reduction of transmission due to contact precautions in infected individuals (Dless) | 0.500 | (0.40, 0.60) | [20,21] |
| $h_2$ | Bacterial clearance rate due to treatment for asymptomatic patients or treatment duration (Days$^{-1}$) | NA[c] | (0.08, 0.12)[b] | [12] Simulated intervention |
| $\sigma_2$ | Effective fraction of asymptomatic carriers clearing colonization following treatment (Dless) | NA[c] | (0.5, 1.0) | Simulated intervention |
| $\gamma_2$ | Reduction constant of transmission due to contact precautions in asymptomatic carriers (Dless) | NA[c] | (0.5, 1.0) | Simulated intervention |
| $f_2$ | The proportion of asymptomatic carrier admissions that are immediately diagnosed as carriers | NA[c] | (0, 1) | Simulated intervention |

[a]Values were derived by determining the proportion of CDI cases in our dataset classified as community-associated CDI. This classification is based on whether or not the time between a CDI diagnosis and the corresponding hospital admission exceeds a time threshold. The baseline time threshold is 3 days, while the range is 1–5 days. [b] In the absence of guidelines, we applied the same range of treatment durations used for CDI patients on colonized individuals. [c] Parameters related to asymptomatic carrier interventions were not included in the baseline scenario, in which no control measures target this population. Parameter ranges were used in the intervention analysis to evaluate the impact of simulated treatment and/or decolonization strategies (see Methods). Abbreviations: CDI, *Clostridioides difficile* infection; CA-CDI, community-associated CDI; HCA-CDI, healthcare-associated CDI; Dless, dimensionless; NA, not applicable.

$$V = \begin{bmatrix} v + \Psi_3 & 0 & 0 & 0 & 0 & 0 \\ 0 & \Psi_3 & 0 & 0 & 0 & -(1-\sigma_1)h_1 \\ 0 & 0 & \Psi_3 + \sigma_2 h_2 + v & 0 & 0 & 0 \\ 0 & 0 & 0 & \Psi_3 + \sigma_2 h_2 & 0 & 0 \\ -v & 0 & 0 & 0 & \Psi_4 + f_1 & 0 \\ 0 & 0 & -v & 0 & -f_1 & (1-\sigma_1)h_1 + \sigma_1 h_1 + \Psi_4 \end{bmatrix}$$

These derivatives were evaluated at the disease-free equilibrium, where all infectious compartments are set to zero and susceptible patients are held at their equilibrium value (S = S'):

$$S' = \frac{\lambda z \left( \Psi_1 + \alpha \right) + \alpha \lambda \omega}{\Psi_2 \left( \Psi_1 + \alpha \right)},$$

Since this approach focuses on intrinsic self-sustaining transmission to the hospital, rates representing the importation of colonized or infected individuals were not included. The Next-Generation Matrix is then defined as:

$$K = FV^{-1},$$

Where the dominant eigenvalue or spectral radius of K corresponds to the intrinsic reproduction number. MATLAB (version R2023a) was used to derive the $R_i$ equation.

**Estimation of colonization amplification index.** The amplification index, defined as the ratio of hospital-discharged colonized patients per day to colonized patients at admission under equilibrium conditions, was calculated from the final time point of the simulation. Although the model reached stability early, we conservatively used the last simulated day, when no appreciable changes in the colonization compartments were observed. The total number of colonized individuals in compartments E, C, $K_1$, and $K_2$ was multiplied by the discharge rate for colonized patients $\Psi_3$ to estimate the daily number of colonized discharges. This was then divided by the expected daily number of colonized admissions, calculated as $\lambda$ times *n*:

$$Ai = \frac{\Psi_3 \left( E + C + K_1 + K_2 \right)}{\lambda n},$$

This metric captures the hospital's contribution to colonization dynamics and complements the $R_i$, which reflects transmission intensity. Values of $A_i > 1$ indicate amplification of colonization during hospitalization. Meanwhile, the symptomatic cases are tracked by the overall incidence of CDI.

**Sensitivity analyses.** Partial rank correlation coefficients (PRCC) were used to assess the sensitivity of model outputs to variation in input parameters. As a standard method in global sensitivity analysis, PRCC quantifies monotonic associations between parameters and outcomes, including $R_i$ and $A_i$, while controlling for the effects of other variables [24]. This method ranks inputs and outputs and uses multiple regression to estimate the independent contribution of each parameter to the outcome. We conducted a univariate sensitivity analysis by expanding the plausible ranges of high-uncertainty parameters and performing simulations to evaluate changes in model outcomes.

**Impact of temporal definitions on healthcare-associated cases.** The heuristic classification of CA-CDI directly affects the number of cases classified as HCA-CDI, influencing estimates of in-hospital transmission. The classification of CA-CDI is based on whether the time from hospital admission to symptom onset of CDI is shorter than a specific time threshold. This threshold is typically set at ≤3 days for disease reporting standards, but that does not perfectly align with which cases originated in the community [16]. Our model examined how varying this threshold from 1 to 5 days impacted the estimation of $R_i$ and $A_i$. We used the observed clinical data for each threshold to estimate the proportion of patients admitted with CA-CDI, π, and thus the corresponding number of cases attributed to in-hospital acquisition. The presumptive number of HCA-CDI was then used to estimate the transmission coefficient, δ, and the resulting $R_i$ and $A_i$ metrics.

**Infection control interventions analysis.** We extended our baseline transmission model by incorporating additional compartments representing control strategies targeting asymptomatic carriers. Given the uncertainty surrounding poorly defined parameters for colonized individuals, including $\gamma_2$, $h_2$, and $\sigma_2$, representing intervention effectiveness and treatment rate, we used random sampling from uniform distributions to explore plausible values.

We evaluated how prophylactic decolonization and contact precautions for asymptomatic carriers affect CDI incidence and the $A_i$ values. Contact precautions aim to limit spread through direct contact [25], while decolonization seeks to eliminate the bacteria, potentially reducing the risk of disease progression and onward transmission. We simulated three scenarios: (1) prophylactic treatment alone, (2) contact precautions alone, and (3) prophylactic treatment combined with contact precautions. For scenario 1, the effectiveness of contact precautions in compartments $K_{1-2}$ was set to zero ($\gamma_2 = 0$), while the treatment completion rate ($h_2$) and effectiveness ($\sigma_2$) were set to the values provided in Table 1. For scenario 2, where contact precautions were simulated, isolation effectiveness ($\gamma_2$) was set to established values, and the treatment completion rate was set to zero ($h_2 = 0$). Lastly, for scenario 3, both interventions were implemented, with the treatment completion rate ($h_2$), treatment effectiveness ($\sigma_2$), and isolation effectiveness ($\gamma_2$) set to their respective values. Comparisons reflect distributions of expected impact under parameter uncertainty rather than paired efficacy.

## Results

During the study period, our healthcare facility recorded a total of 43,320 admissions, with an average of 76 admissions per day. Based on laboratory tests conducted, 4,767 episodes of diarrhea were identified; of these, 659 (13.8%) episodes were confirmed as CDI by PCR. Among the confirmed cases, 10 (1.5%) episodes were considered duplicated due to testing intervals of less than 7 days, and 10 (1.5%) episodes were outpatients, resulting in a final dataset of 639 CDI cases.

According to our records, the overall incidence of CDI was 14.8 cases per 1,000 admissions. Based on the standard case definition specifying that all CA-CDI cases are diagnosed within 3 days of admission, the HCA-CDI incidence was 10.2 cases per 1,000 admissions, while the CA-CDI incidence was 4.6 cases per 1,000 admissions, with 441 and 198 cases reported, respectively.

Across 1,000 sets of parameters and the standard three-day cutoff for HCA-CDI versus CA-CDI classification, the median $A_i$ was 1.9 (Q1–Q3: 1.7, 2.1), indicating that the number of colonized patients at discharge was approximately 90% higher than at admission. The median $R_i$ was 0.61 (Q1–Q3: 0.53, 0.71), with $R_i < 1$ in 99.8% of all parameter sets, indicating low potential for self-sustaining transmission within the hospital (Fig 2). We observed a positive correlation between $R_i$ and $A_i$ (S1 Fig).

### Sensitivity analyses

According to the global sensitivity analysis, the transmission rate of symptomatic patients, the relative transmissibility of colonized individuals, the antibiotic usage rate, and the fraction of susceptible patients admitted were the strongest positive predictors of both $R_i$ and $A_i$ estimates. Conversely, the discharge rates for nonsusceptible patients, asymptomatic carriers, and susceptible patients demonstrated the strongest negative correlations (S1 and S2 Tables). In the univariate analysis, we observed similar qualitative behavior of a subcritical intrinsic reproduction number and positive colonization amplification for most scenarios. Only scenarios with low values of the fraction of colonized individuals who develop symptoms during hospitalization ($\varepsilon$), low colonization prevalence at admission ($n$), and with prolonged incubation periods ($v$) ($\geq 9$ days) showed $R_i > 1$ (S3 Table). Within the explored parameter space, we observed monotonic effects on outcomes (S3 Table).

### Temporal definitions on healthcare-associated cases

Our sensitivity analysis explored how the time threshold for distinguishing between CA-CDI and HCA-CDI impacted $R_i$ and $A_i$ estimates (Fig 3). The median reproduction number increased from 0.56 to 0.74 as the time threshold decreased from 5 days to 1 day. Even under the most inclusive definition (1-day threshold), which classified more cases as hospital-acquired, $R_i$ remained below one in 95.3% of simulations, suggesting limited potential for sustained transmission. On the other hand, the $A_i$ ranged from 1.80 to 2.10 as the time threshold decreased from 5 days to 1 day.

## a) Intrinsic reproduction number

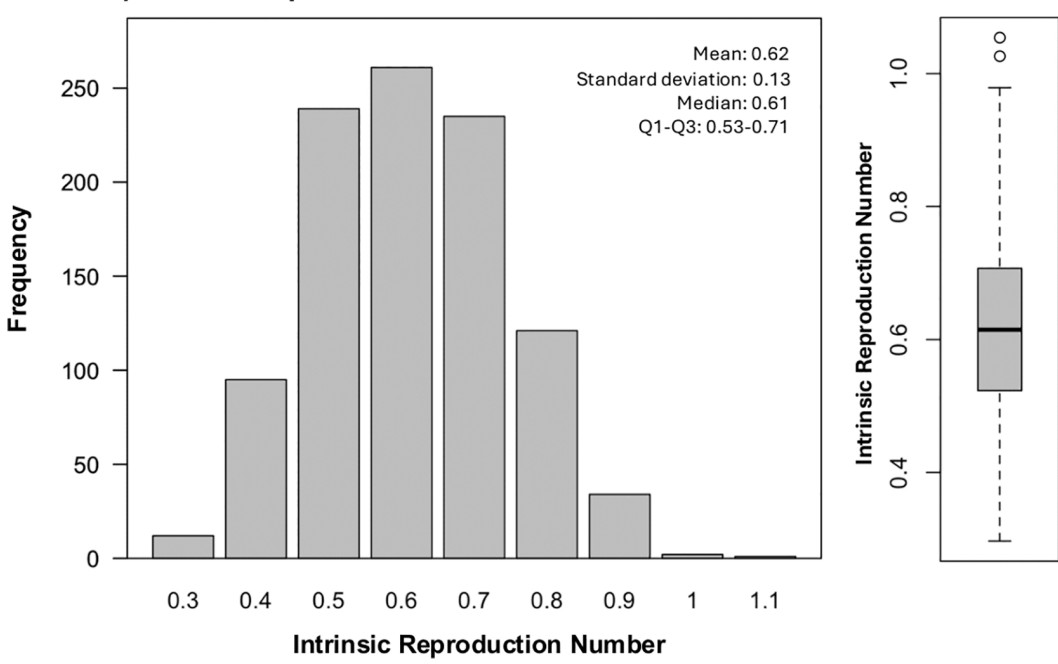

## b) Amplification index

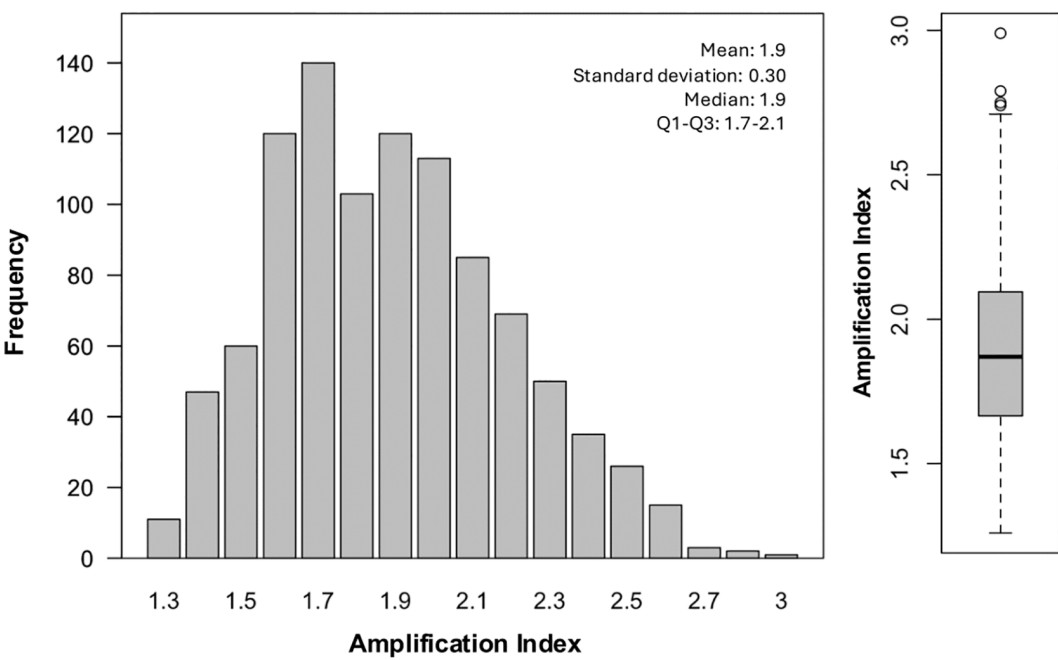

**Fig 2. Distribution of the intrinsic reproduction number (R$_i$) and the amplification index (A$_i$) values.** The (a) distribution and box plot of the intrinsic reproduction number, and (b) the colonization amplification index, for 1,000 randomly sampled parameter values. A cutoff of 3 days was used to distinguish healthcare-associated *Clostridioides difficile* infection (HCA-CDI) from community-associated *Clostridioides difficile* infection (CA-CDI). In box plots, the center line represents the median, the box represents the interquartile range (Q1–Q3), the whiskers extend to the most extreme values within 1.5 × the interquartile range, and points represent outliers.

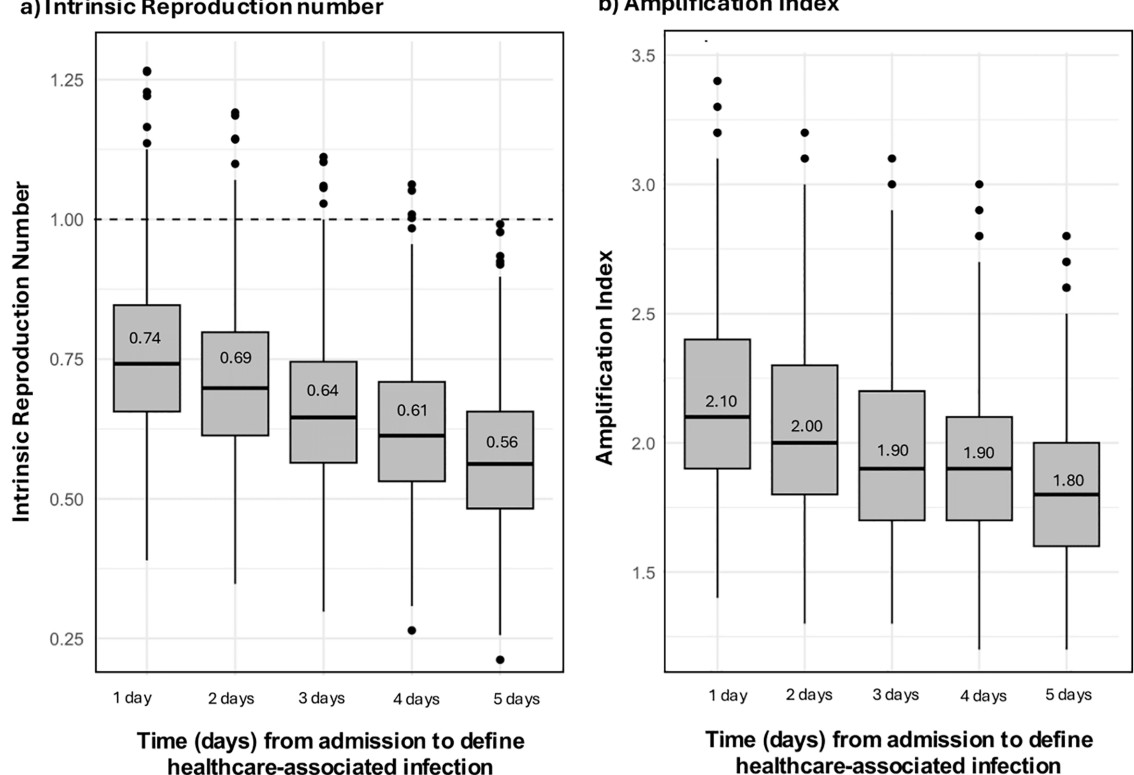

**Fig 3. Analysis of temporal definitions of healthcare-associated cases.** Boxplots of (a) the intrinsic reproduction number ($R_i$) and (b) the amplification index ($A_i$) under alternative temporal thresholds used to classify healthcare-associated infection. In box plots, the center line represents the median, the box represents the interquartile range (Q1–Q3), the whiskers extend to the most extreme values within 1.5 × the interquartile range, and points represent outliers.

Compared with the baseline definition of HCA-CDI (>3 days post-admission), $A_i$ and $R_i$ increased by 11% and 21%, respectively, when a threshold of ≥1 day post-admission was used to define HCA-CDI. Conversely, using a threshold of ≥5 days post-admission to define hospital-acquired disease decreased the baseline values by 5% and 8%, respectively (Fig 3).

## Infection control interventions analysis

To assess the impact of contact precautions and prophylactic treatment for asymptomatic carriers, we simulated three intervention strategies: treatment of carriers, isolation of carriers, and both combined (Fig 4). Our results depended on the proportion of patients with asymptomatic carriage who were diagnosed on admission ($f_2$) and the effectiveness parameters for isolation ($\gamma_2$) and decolonization ($\sigma_2$). As the effectiveness of interventions improves, $A_i$ and the overall CDI incidence decrease.

When implemented individually, contact precautions were consistently more effective at reducing $A_i$ and CDI incidence than prophylactic decolonization. Notably, the combined application of both interventions produced the largest effect. For example, isolating and treating 20% of colonized individuals at admission reduced the amplification index by ~7.5% and CDI incidence by 3.6%. Combining both interventions yielded an approximately additive reduction in CDI incidence, with the combined effect roughly equal to the sum of their individual effects. However, the decrease in $A_i$ was less pronounced, although still approximately additive.

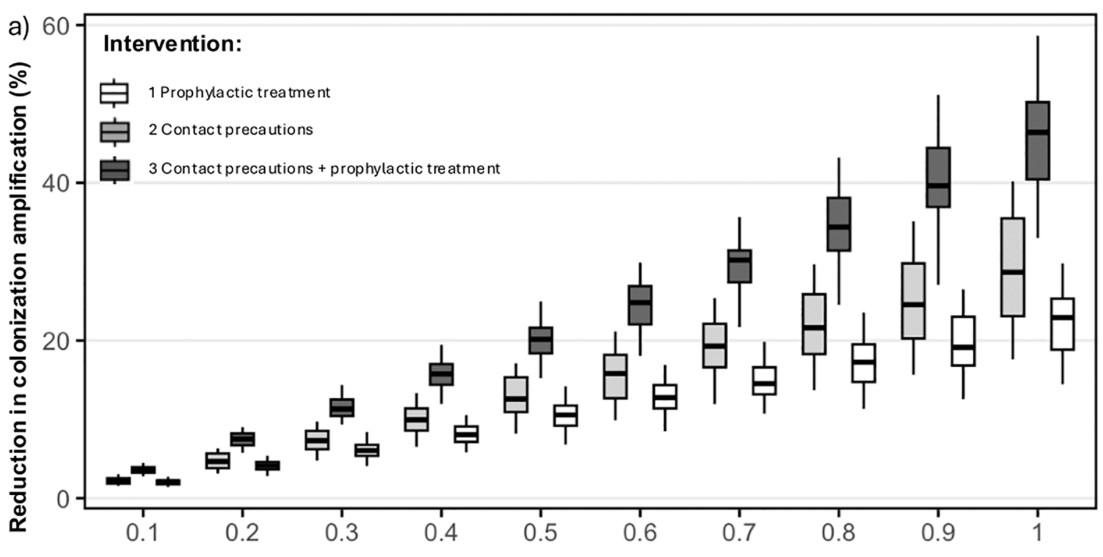

| f₂ value | 0.1 | 0.2 | 0.3 | 0.4 | 0.5 | 0.6 | 0.7 | 0.8 | 0.9 | 1.0 |
|---|---|---|---|---|---|---|---|---|---|---|
| Reduction (%) Intervention 1 | 2.0 | 4.1 | 6.1 | 8.1 | 10.5 | 12.9 | 14.8 | 17.1 | 19.7 | 22.1 |
| Reduction (%) Intervention 2 | 2.2 | 4.7 | 7.3 | 9.9 | 12.8 | 15.5 | 19.1 | 22.1 | 24.8 | 28.8 |
| Reduction (%) Intervention 3 | 3.6 | 7.5 | 11.5 | 15.7 | 20.0 | 24.4 | 29.5 | 34.4 | 40.3 | 45.5 |

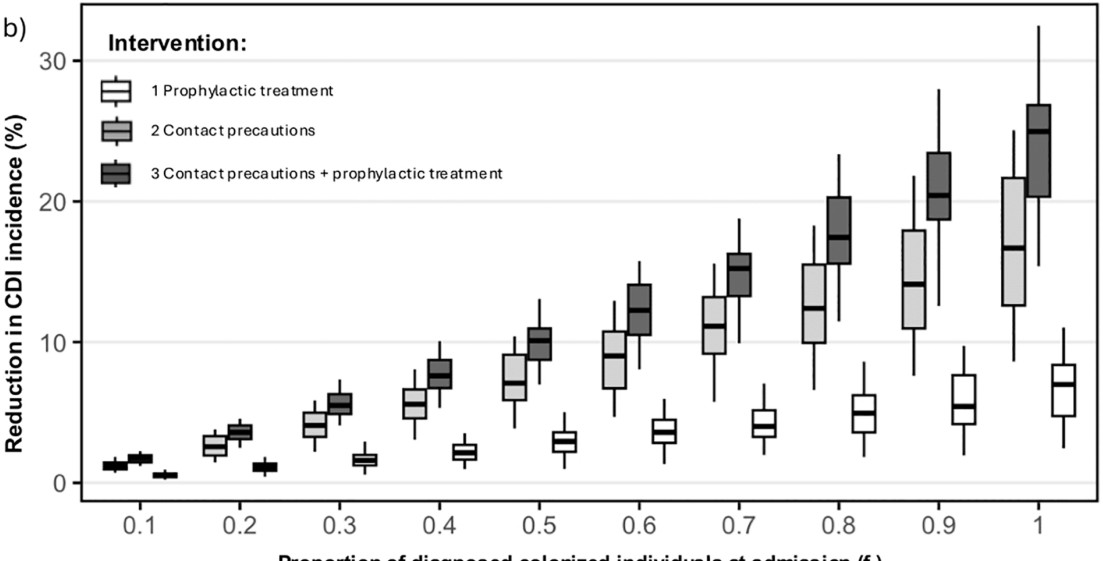

| f₂ value | 0.1 | 0.2 | 0.3 | 0.4 | 0.5 | 0.6 | 0.7 | 0.8 | 0.9 | 1.0 |
|---|---|---|---|---|---|---|---|---|---|---|
| Reduction (%) Intervention 1 | 0.5 | 1.1 | 1.6 | 2.2 | 2.9 | 3.7 | 4.2 | 4.9 | 5.8 | 6.6 |
| Reduction (%) Intervention 2 | 1.2 | 2.6 | 4.1 | 5.5 | 7.3 | 8.8 | 11.0 | 12.7 | 14.3 | 16.8 |
| Reduction (%) Intervention 3 | 1.7 | 3.6 | 5.6 | 7.7 | 9.9 | 12.2 | 14.9 | 17.7 | 20.9 | 23.9 |

**Fig 4. Impact of interventions focused on asymptomatic carriers of *C. difficile*.** Changes in the colonization amplification index (top) and CDI incidence (bottom) when applying: 1) prophylactic treatment of carriers at the time of admission, 2) contact precautions for carriers at the time of admission, and 3) prophylactic treatment combined with contact precautions. The x-axis shows the proportion of diagnosed colonized individuals at admission (f₂).

In box plots, the center line represents the median, the box represents the interquartile range (Q1–Q3), the whiskers extend to the most extreme values within 1.5 × the interquartile range, and points represent outliers.

## Discussion

In this study, we developed a compartmental model that incorporated both symptomatic and asymptomatic individuals to quantify *C. difficile* transmission within the hospital setting. Using this framework, we estimated two complementary metrics, $R_i$ and $A_i$, to capture both in-hospital transmission and the hospital's contribution to community colonization. We further explored the robustness of these estimates through sensitivity analyses of model parameters, assessed the influence of varying case attribution definitions, and evaluated targeted interventions on asymptomatic carriers.

Our findings highlight the bidirectional relationship between hospital and community *C. difficile* burden. Across model scenarios, $R_i$ remained consistently below 1, suggesting that in-hospital transmission alone does not sustain endemicity. Instead, ongoing importation of cases from the community appears necessary to maintain endemicity within healthcare institutions. This highlights the importance of accounting for internal and external dynamics in infection control strategies [26,27].

Hospitals contribute to the broader community burden by discharging colonized individuals, many of whom remain undetected by current screening practices. By introducing $A_i$, our analysis indicates a consistent amplification of colonized cases within the healthcare setting, even in scenarios where $R_i$ remains low. This pattern is explained by the persistence of colonization among asymptomatic patients until discharge. This hospital-driven amplification may indirectly sustain healthcare-associated transmission if colonized individuals are later readmitted.

Our findings align with and complement previous reports documenting increased colonization and higher risk of CDI among individuals recently exposed to healthcare settings [18,26,28–30]. Our estimates of $A_i$ are sensitive to parameter uncertainty, center-specific characteristics, and stochastic fluctuations. These results should therefore be interpreted as a mean-field estimate, highlighting the important role of unobserved hospitalized colonized patients for the overall *C. difficile* burden in both hospital and community settings.

The sensitivity of $R_i$ and $A_i$ to heuristic classification of HCA-CDI cases based on the timing of CDI diagnosis underscores how standardized definitions can significantly influence transmission estimates. More accurate classification of cases and assessment of healthcare-associated transmission requires consideration of the stochastic nature of disease progression.

Sensitivity analyses exploring a wide range of parameters suggest that $R_i < 1$ for essentially all tested parameter values, with $R_i > 1$ observed only in a restricted set of values reflecting reduced importation of colonized patients or lower progression to symptomatic infection, conditions that require higher transmission to reproduce the observed incidence of symptomatic cases. Although such scenarios are plausible, they are less consistent with empirical observations, given that colonization at admission typically exceeds 3% and incubation periods are most commonly shorter than one week [12,17,31]. Evidence on progression from colonization to symptomatic disease is limited due to the scarcity of prospective studies with systematic follow-up; however, available data suggest that this fraction is likely greater than 6% [18].

Unlike previous studies, our model explicitly assesses colonization amplification and incorporates interventions targeting asymptomatic carriers. This underscores the need to expand strategies beyond managing symptomatic cases to include interventions targeting asymptomatic carriers at the time of admission and, potentially, at discharge. From a policy perspective, our results suggest that relying solely on $R_i$ to evaluate hospital performance may underestimate the broader epidemiological impact of healthcare institutions on *C. difficile* transmission.

Given evidence that asymptomatic carriers may shed *C. difficile* at levels comparable to symptomatic patients, interventions targeting this group, such as contact precautions and antimicrobial prophylaxis, have been proposed as potential strategies to reduce onward transmission [13,14,16]. Our model showed that both contact precautions and prophylactic treatment targeting asymptomatic carriers reduced $A_i$ and infection incidence. Contact precautions consistently

outperformed prophylaxis, likely due to their immediate effect on interrupting transmission, whereas prophylactic treatment requires time to achieve bacterial clearance, delaying its impact on transmission dynamics. However, unlike prophylaxis, contact precautions do not affect the risk of progression from colonization to symptomatic disease.

Although quasi-experimental studies have supported the effectiveness of contact precautions [32], strategies to prevent the progression from asymptomatic carriage to active infection may be particularly valuable in settings where transmission from asymptomatic carriers is not the primary driver of spread. For example, Miles-Jay and colleagues showed that in intensive care units (ICUs), only 1% of patients who tested negative for *C. difficile* upon admission acquired the pathogen through cross-transmission, with most cases occurring in previously colonized patients. However, it is essential to note that conditions in ICUs may not reflect those in general hospital wards. ICU patients were typically housed in single rooms and benefited from daily cleaning with sporicidal disinfectants, which likely reduced opportunities for cross-transmission compared to other healthcare settings [33].

There are practical concerns to implementing contact precautions for carriers, including higher costs, potential impacts on patient well-being, and ethical considerations around isolating individuals without symptoms [34]. Although the effectiveness of contact precautions depends on implementation and compliance, current infection control frameworks provide the infrastructure for implementation. Our model can be helpful for cost-effectiveness studies of these interventions by providing estimates of CDI cases and carriers averted.

The relative effectiveness of prophylactic treatment is tempered by studies showing success rates as low as 30%–40% in asymptomatic and symptomatic patients [19,35]. Meanwhile, complex microbiome interactions and heterogeneous bacterial clearance rates associated with antibiotic use were not incorporated into our framework and may modify the actual impact of these interventions. Moreover, prophylactic use of antibiotics may carry unintended consequences, including increased risk of secondary infections such as vancomycin-resistant enterococci, thus contributing to the broader challenge of antimicrobial resistance [35,36]. As new decolonization strategies emerge and our understanding of microbiome dynamics improves, our model can be used to ascertain whether targeted or universal decolonization becomes a preferred strategy.

This study has several limitations. First, the biological mechanisms driving colonization and progression to symptomatic infection remain poorly understood, limiting our ability to model these transitions precisely. We quantified parameter uncertainty but did not quantify uncertainty arising from alternative model structures. Second, a compartmental model approach may overlook heterogeneity within populations and individual variations in movement rates. Instead, this approach relies on assumptions of homogeneous mixing and uniform transition rates between compartments. Third, the data used to calibrate the model included a period during the COVID-19 pandemic, which could alter transmission dynamics. Prospective studies with repeated testing, genomic sequencing, and expanded screening of healthcare workers could substantially improve parameter estimates and characterization of infection dynamics. In addition, these types of studies can improve model validation as model predictions can be tested prospectively.

Lastly, the generalizability of our findings should be considered with caution, as specific estimates of $R_i$ and $A_i$ may vary across institutions depending on local patient demographics, sources of admission, infection control practices, and patterns of antibiotic use. Our model structure captures fundamental and generalizable epidemiological processes underlying *C. difficile* transmission that are applicable to other healthcare settings and can be adapted with relatively minor parameter changes or compartmental extensions under parsimonious scenarios that preserve interpretability. While the aforementioned limitations inspire opportunities for future work, our model provides insight for assessing the transmission dynamics and control of *C. difficile* in healthcare settings.

## Supporting information

**S1 Fig. Relationship between intrinsic reproduction number and colonization amplification index.** Scatter plot showing the relationship between $R_i$ and $A_i$ across 1,000 simulations with randomly sampled parameter values. Each point represents the outcome from a single parameter set.
(DOCX)

**S1 Table. Partial rank correlation coefficients for input parameters in the intrinsic reproduction number estimates.** Partial rank correlation coefficients (PRCC) were obtained after Monte Carlo sampling of parameter values. The PRCC provides adjusted correlation values between model parameters and the intrinsic reproduction number. A cutoff of 3 days was used to distinguish community-associated (CA-CDI) from healthcare-associated CDI (HCA-CDI).
(DOCX)

**S2 Table. Partial rank correlation coefficients for input parameters in the colonization amplification index estimates.** Partial rank correlation coefficients (PRCC) for the colonization amplification index were obtained after Monte Carlo sampling of parameter values. A cutoff of 3 days was used to distinguish CA-CDI from HCA-CDI.
(DOCX)

**S3 Table. Univariate sensitivity analysis of model parameters and their impact on the intrinsic reproduction number and the colonization amplification index.** Each parameter was varied individually across a predefined range while all other parameters were held at their baseline values. The resulting values of the intrinsic reproduction number and the colonization amplification index are shown for each parameter value.
(DOCX)

## Author contributions

**Conceptualization:** Daniel De-la-Rosa-Martinez, Travis C. Porco, Ashley Hazel, Seth Blumberg.

**Formal analysis:** Daniel De-la-Rosa-Martinez, Travis C. Porco, Karim Khader, Seth Blumberg.

**Investigation:** Daniel De-la-Rosa-Martinez, Ashley Hazel, Xinran Liu, Seth Blumberg.

**Methodology:** Daniel De-la-Rosa-Martinez, Travis C. Porco, Xinran Liu, Karim Khader, Seth Blumberg.

**Resources:** Travis C. Porco, Seth Blumberg.

**Software:** Daniel De-la-Rosa-Martinez, Travis C. Porco.

**Supervision:** Travis C. Porco, Karim Khader, Seth Blumberg.

**Validation:** Daniel De-la-Rosa-Martinez.

**Visualization:** Daniel De-la-Rosa-Martinez, Travis C. Porco, Seth Blumberg.

**Writing – original draft:** Daniel De-la-Rosa-Martinez, Seth Blumberg.

**Writing – review & editing:** Daniel De-la-Rosa-Martinez, Travis C. Porco, Ashley Hazel, Xinran Liu, Karim Khader, Seth Blumberg.

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
