## [Editor Report · Decision Letter 0]

17 Jul 2025

Dear Dr De-la-Rosa-Martinez,

Thank you for submitting your manuscript entitled "Colonization Amplification despite Limited In-Hospital Transmission: Modeling the Epidemiological Paradox of C. difficile and the Impact of Control Strategies in Healthcare Settings." for consideration by PLOS Medicine.

Your manuscript has now been evaluated by the PLOS Medicine editorial staff as well as by an academic editor with relevant expertise and I am writing to let you know that we would like to send your submission out for external peer review.

**Please also revise the title and abstract as per PLOS Medicine specifications. Please also ensure that include a data availability statement, deposit the code in a public repository, complete the acknowledgments section of your manuscript and clarify if ethics were waived. Please identify the institution in the manuscript from which the data were obtained and discuss the study limitations. In particular we ask you to address the issue of generalizability--are the findings likely to be specific to your institution, or to institutions in the US? Please explain to the general reader the policy or clinical impact of your findings and the advance over existing related studies, including reference 26 and doi: 10.1038/s41591-023-02549-4, and discuss whether there are divergent views in terms of how healthcare-associated CDIs are sustained.

For clinical studies, please upload a copy of your trial study protocol as a supporting information file. The study protocol should be the version submitted for approval to the institutional review board or ethics committee, should include any amendments to the study protocol, as well as the date of their approval by the institutional review or ethics committee. Please also detail any deviations from the study protocol in the Methods section of your manuscript. The editors will consider the protocol and study conduct prior to a final decision for external review.

Please re-submit your manuscript within two working days, i.e. by Jul 21 2025 11:59PM.

Kind regards,

Alison Farrell, Ph.D.

Senior Editor

PLOS Medicine

---

## [Decision Letter · Decision Letter 1]

19 Dec 2025

Dear Dr De-la-Rosa-Martinez,

Many thanks for submitting your manuscript "Colonization Amplification despite Limited In-Hospital Transmission: Modeling the Epidemiological Paradox of C. difficile and the Impact of Control Strategies in Healthcare Settings." (PMEDICINE-D-25-02546R1) to PLOS Medicine. We apologize for the delay in conveying to you our decision. The paper has now been reviewed by subject experts and a statistician; their comments are included below and can also be accessed here: [LINK]

As you will see, the reviewers find the work interesting and have made suggestions to improve clarity and strength of the findings. After discussing the paper with the editorial team I'm pleased to invite you to revise the paper in response to the reviewers' comments. We plan to send the revised paper to some or all of the original reviewers, and we cannot provide any guarantees at this stage regarding publication.

We ask that you submit your revision by Jan 09 2026 11:59PM. However, if this deadline is not feasible, please contact me by email, and we can discuss a suitable alternative.

Don't hesitate to contact me directly with any questions (afarrell@plos.org).

Best regards,

Alison

Alison Farrell, Ph.D.

Senior Editor

PLOS Medicine

afarrell@plos.org

Comments from the reviewers:

Reviewer #1: The authors present a thought-provoking piece of modeling work based in part on real-world clinical data which calls into question some of the assumptions underpinning current C difficile prevention measures. Their modeling approach is well explained (even for a clinician and amateur modeler like me) and assumed input values well delineated (Table 1 being particularly helpful).

I do have a few questions mostly related to curiosity about how the modeled results can be compared to some of the limited real world studies looking at similar numbers:

1) With the Ai of 1.9 - suggesting a 90% increase in colonization by discharge relative to admission: limited prospective surveys (such as that by Dubberke et al AAC 2015; https://journals.asm.org/doi/10.1128/aac.00642-15) do confirm increases in colonization rates between admission and discharge, but rates in this prospective microbiologic survey at least seem much lower than the 90% increase here. How might the authors reconcile this estimate with existing survey data? Is there perhaps also a time-dependent component? (e.g., rather than an overall average amplification in colonization, perhaps a per day hospitalized increase?)

2) Aside from suggesting that studies of isolation or prophylactic treatment of asymptomatically colonized patients, might the authors suggest any other prospective epidemiologic studies which might provide real empirical estimates to inform model relevance/accuracy?

3) Besides just the hospital component, are the authors able to account for potential roles for nursing facilities as these can serve as a source of admission, a site for discharge, and may be associated with high carriage rates as well? OK if unable - just curious.

Reviewer #2: This manuscript uses an epidemiological model to explore the dynamics of C. diff transmission using data from a large academic hospital. They introduce a metric to measure the amplification of host numbers between admission and exit. I am not an expert on C. diff and, as such, cannot discuss prior work in this area, so I will focus my comments on the technical aspects of the paper. This work is valuable and I am always happy to see the impact of nosocomial infection on community dynamics being investigated. In general, I think the approach applied in this work is strong, but before I can recommend publication, I had some technical questions for the authors.

222/223 - parameters were sampled from uniform distributions +/- 20% of their baseline values - You mention on lines 305/306 that some parameters are particularly poorly characterised. Given this, is a consistent choice of 20% across parameters justified, or should you widen the uncertainty bounds for some parameters and reduce them for others, conditional on the degree of uncertainty of the parameters of interest? I worry that your implied uncertainty distribution may very poorly describe the real uncertainty given this.

A_i - You calculate A_i directly from the model, assuming that stability has been reached. How did you set the initial conditions for your compartments? And, if you set them semi-arbitrarily, did you check that there aren't multiple equilibria with the infection present for your models? (If there's only one, you might be able to get an analytic expression for A_i in terms of your paramters, which would be nice.)

As for a couple of non-technical suggestions that I think would improve the paper:

1. I'd be very interested to see a scatter graph of R_i against A_i to see how strongly correlated they are.

2. On that note, some discussion of how the amplification is occuring while R_i is less than 1 around 405 would be useful to help explain the concept of what's going on. I'm assuming that it's that while less than 1 person is getting infected on average, the initial person remains infected, so you end up on average with more infections leaving than entered. Is that what's happening?

Minor pdf-conversion(?) notes

211/207 The parameter multiplying the E compartment isn't rendering

247/248 Any and all symbols to the left of (x) on these lines aren't rendering

Reviewer #3: This is an infectious disease modelling study that examined the role of healthcare associated C. difficile infection in association with the broader community to solve the 'epidemiological paradox'. The paper is well written and clear. The description of method is clear and in sufficient detail for replication. The inclusion of several sensitivity analyses are useful in understand how the model output could change by different assumptions / definitions. I only have several minor comments:

Line 100: missing '2)' before Suspectable (S)...

Lines 205-212: The use of squared brackets confuses the with in-text citation of references.

Lines 284-289: While PRCC it is a standard method in global sensitivity analysis, it does not capture any non-monotonic effect of parameters on outputs. Could the author discuss if effect non-monotonic is possible?

---

* Please upload any figures associated with your paper as individual TIF or EPS files with 300dpi resolution at resubmission; please read our figure guidelines for more information on our requirements: http://journals.plos.org/plosmedicine/s/figures. While revising your submission, we strongly recommend that you use PLOS's NAAS tool (https://ngplosjournals.pagemajik.ai/artanalysis) to test your figure files. NAAS can convert your figure files to the TIFF file type and meet basic requirements (such as print size, resolution), or provide you with a report on issues that do not meet our requirements and that NAAS cannot fix.

After uploading your figures to PLOS's NAAS tool - https://ngplosjournals.pagemajik.ai/artanalysis, NAAS will process the files provided and display the results in the "Uploaded Files" section of the page as the processing is complete.

If the uploaded figures meet our requirements (or NAAS is able to fix the files to meet our requirements), the figure will be marked as "fixed" above. If NAAS is unable to fix the files, a red "failed" label will appear above.

When NAAS has confirmed that the figure files meet our requirements, please download the file via the download option, and include these NAAS processed figure files when submitting your revised manuscript.

FIGURES AND TABLES

SUPPLEMENTARY MATERIAL

REFERENCES

* Please add URL(s) for funders.

* Please ensure you include a discussion of the study limitations with specific reference to a lack of sequence data.

* Please discuss methods to independently validate the model.

* Please acknowledge individuals and UCSF staff who collected the data used in your modeling.

MODELLING STUDIES

The following list is derived from Geoffrey P Garnett, Simon Cousens, Timothy B Hallett, Richard Steketee, Neff Walker. Mathematical models in the evaluation of health programmes. (2011) Lancet DOI:10.1016/S0140-6736(10)61505-X:

* If pertinent, please provide a diagram that shows the model structure, including how the natural history of the disease is represented, the process and determinants of disease acquisition, and how the putative intervention could affect the system.

* Please provide a complete list of model parameters, including clear and precise descriptions of the meaning of each parameter, together with the values or ranges for each, with justification or the primary source cited and important caveats about the use of these values noted.

* Please provide a clear statement about how the model was fitted to the data, including goodness-of-fit measure, the numerical algorithm used, which parameter varied, constraints imposed on parameter values, and starting conditions.

* For uncertainty analyses, please state the sources of uncertainties quantified and not quantified [can include parameter, data, and model structure].

* Please provide sensitivity analyses to identify which parameter values are most important in the model. Uncertainty estimates seek to derive a range of credible results on the basis of an exploration of the range of reasonable parameter values. The choice of method should be presented and justified.

* Please discuss the scientific rationale for the choice of model structure and identify points where this choice could influence conclusions drawn. Please also describe the strength of the scientific basis underlying the key model assumptions.

* For studies that develop a prediction model or evaluate its performance, please ensure that the study is reported according to the TRIPOD statement (https://www.equator-network.org/reporting-guidelines/tripod-statement) and include the completed checklist as Supporting Information. Please add the following statement, or similar, to the Methods: "This study is reported as per the Transparent Reporting of a Multivariable Prediction Model for Individual Prognosis Or Diagnosis (TRIPOD) statement (S1 Checklist)." For studies using machine learning, please use the TRIPOD-AI checklist. When completing the checklist, please use section and paragraph numbers, rather than page numbers.

---

## [Decision Letter · Decision Letter 2]

3 Mar 2026

Dear Dr. De-la-Rosa-Martinez,

Thank you very much for re-submitting your manuscript "Colonization Amplification despite Limited In-Hospital Transmission: Modeling the Epidemiological Paradox of C. difficile and the Impact of Control Strategies in Healthcare Settings." (PMEDICINE-D-25-02546R2) for review by PLOS Medicine.

I have discussed the paper with my colleagues and the academic editor and it was also seen again by 3 reviewers. I am pleased to say that provided the remaining editorial and production issues are dealt with we are planning to accept the paper for publication in the journal.

[LINK]

We look forward to receiving the revised manuscript by Mar 10 2026 11:59PM.

Sincerely,

Alison Farrell, Ph.D.

Senior Editor

PLOS Medicine

plosmedicine.org

Requests from Editors:

Title needs to adhere to PLOS Medicine criteria. Please confirm that your title complies with PLOS Medicine's style. Your title must be nondeclarative and not a question. It should begin with main concept if possible. "Effect of" should be used only if causality can be inferred, i.e., for an RCT. Please place the study design ("A randomized controlled trial," "A retrospective study," "A modelling study," etc.) in the subtitle (ie, after a colon).

C. difficile should be spelled out in full at first use in title, line 26 of Abstract, in the Author Summary, and in Introduction.

* Please confirm that all numbers presented in the abstract are present and identical to numbers presented in the main manuscript text.

Use commas for ranges, not hyphens.

Line 37 ‘Ai’—i should be subscripted

Line 41: CDI needs to be defined at first use.

Line49: “which contributes to’—please temper claim as community spread is not directly addressed in this study.

Please ensure that the Abstract provides all the information relevant to this study type https://journals.plos.org/plosmedicine/s/submission-guidelines#loc-abstract

Line 60: change ‘diseased’ to ‘disease’

In the Author Summary, please limit each section to ~3 bullet points and re-write in an accessible, non-technical manner for a wide audience that includes both scientists and non-scientists.

Please include the main limitations of the study in non-technical language in the last bullet point of “What do these findings mean?”

* Please ensure that the Introduction ends with a clear description of the study question or hypothesis.

Please define all abbreviations in Tables and Figures.

Please ensure that all Figures and Tables, including Supplementary items, have titles and legends.

Please define all elements of box plots in the figure caption - center line, box limits and whiskers.

Please include a Statistical Analysis subsection in the Methods.

Please clarify in the Methods where the data come from and clarify to editors why ethics approval was required.

If individuals provided feedback on the manuscript, please include them in the Acknowledgments, otherwise remove the section if not persons are to be acknowledged.

Please address the following points derived from Geoffrey P Garnett, Simon Cousens, Timothy B Hallett, Richard Steketee, Neff Walker. Mathematical models in the evaluation of health programmes. (2011) Lancet DOI:10.1016/S0140-6736(10)61505-X:

* Please provide a diagram that shows the model structure, including how the disease natural history is represented, the process and determinants of disease acquisition, and how the putative intervention could affect the system.

* Please provide a complete list of model parameters, including clear and precise descriptions of the meaning of each parameter, together with the values or ranges for each, with justification or the primary source cited, and important caveats about the use of these values noted.

* Please provide a clear statement about how the model was fitted to the data including where relevant goodness-of-fit measure, the numerical algorithm used, which parameter varied, constraints imposed on parameter values, and starting conditions.

* For uncertainty analyses, please state the sources of uncertainties quantified and not quantified this can include parameter, data, and model structure.

* Please provide sensitivity analyses to identify which parameter values are most important in the model. Uncertainty estimates seek to derive a range of credible results on the basis of an exploration of the range of reasonable parameter values. The choice of method should be presented and justified.

* Please discuss the scientific rationale for this choice of model structure and identify points where this choice could influence conclusions drawn. Please also describe the strength of the scientific basis underlying the key model assumptions.

Comments from Reviewers:

Reviewer #1: The authors have answered each of my questions. I have no further suggested edits.

Reviewer #2: Thank you for the extensive work you have done in response to my review. I agree that the technical discussion to my point about equilibria might just add mathematical complexity to the paper for no real gain for the reader, but I am content that the issue has been considered, even if it doesn't make it into the text explicitly. I am now happy to unequivocally recommend publication.

Reviewer #3: The authors have addressed my previous comments satisfactorily. Congratulations on completing this study.

[LINK]

---

## [Editor Report · Decision Letter 3]

26 Mar 2026

Dear Dr De-la-Rosa-Martinez,

On behalf of my colleagues and the Academic Editor, Anthony Bai, I am pleased to inform you that we have agreed to publish your manuscript "Colonization Amplification despite Limited In-Hospital Transmission: A Modeling Study of Clostridioides difficile transmission and Control in Healthcare Settings." (PMEDICINE-D-25-02546R3) in PLOS Medicine.

Please note the following editorial requests at this time:

We ask that you provide a doi (e.g. via Zenodo) for the code.

Please add an author contribution statement.

Please revise the title to reduce repetition and ensure accuracy (as the study derives from one healthcare facility). We suggest: Clostridioides difficile Colonization Amplification despite Limited In-Hospital Transmission: A Modeling Study

PRESS

Sincerely,

Alison Farrell, Ph.D.

Senior Editor

PLOS Medicine